# *In vivo* calcium imaging reveals directional sensitivity of C-low threshold mechanoreceptors

Evangelia Semizoglou[1] , Laure Lo Re[2], Steven J. Middleton[3], Jimena Perez-Sanchez[3] ,
Tommaso Tufarelli[4] , David L. Bennett[3] and Kim I. Chisholm[5]

[1]*Department of Neurology, Brigham & Women's Hospital, Harvard Medical School, Boston, USA*
[2]*Tafalgie Therapeutics, Campus de Luminy, Marseille, France*
[3]*Nuffield Department of Clinical Neurosciences, University of Oxford, Oxford, UK*
[4]*School of Mathematical Sciences, The University of Nottingham, University Park, Nottingham, UK*
[5]*School of Life Sciences, The University of Nottingham, Queen's Medical Centre, Nottingham, UK*

Handling Editors: Katalin Toth & Nathan Schoppa

The peer review history is available in the Supporting Information section of this article
(https://doi.org/10.1113/JP286631#support-information-section).

**The Journal of Physiology**

**Abstract figure legend** This study utilizes TH[CreERT2]GCaMP6 mice, which express GCaMP6 in tyrosine hydroxylase-positive C-LTMRs neurons, to perform *in vivo* calcium imaging. This approach allows for the visualization of intracellular calcium as a proxy for neuronal activity, revealing responses of C-LTMRs to punctate mechanical stimuli and highlighting their directional sensitivity to brush stimuli. Created in BioRender. Chisholm, K. (2024) BioRender.com/g66l177

The Journal of Physiology

**Abstract**   C-low threshold mechanoreceptors (C-LTMRs) in animals (termed C-tactile (CT) fibres in humans) are a subgroup of C-fibre primary afferents, which innervate hairy skin and respond to low-threshold punctate indentations and brush stimuli. These afferents respond to gentle touch stimuli and are implicated in mediating pleasant/affective touch. These afferents have traditionally been studied using low-throughput, technically challenging approaches, including microneurography in humans and teased fibre electrophysiology in other mammals. Here we suggest a new approach to studying genetically labelled C-LTMRs using *in vivo* calcium imaging. We used an automated rotating brush stimulus and von Frey filaments, applied to the hairy skin of anaesthetized mice to mirror light and affective touch. Simultaneously we visualized changes in C-LTMR activity and confirmed that these neurons are sensitive to low-threshold punctate mechanical stimuli and brush stimuli with a strong preference for slow brushing speeds. We also reveal that C-LMTRs are directionally sensitive, showing more activity when brushed against the natural orientation of the hair. We present *in vivo* calcium imaging of genetically labelled C-LTMRs as a useful approach that can reveal new aspects of C-LTMR physiology.

(Received 28 March 2024; accepted after revision 28 November 2024; first published online 15 January 2025)
**Corresponding author** K. Chisholm: Room E153c, Queen's Medical Centre, School of Life Sciences, The University of Nottingham, Nottingham UK.     Email: kim.chisholm@nottingham.ac.uk

### Key points

- C-low threshold mechanoreceptors are sensitive to the directionality of a brush stimulus, being preferentially activated by brushing against the grain of the hair, compared with brushing with the grain of the hair. This is surprising as brushing against the grain of the hair is considered less pleasant.
- *In vivo* calcium imaging is a useful approach to the study of C-low threshold mechanoreceptors.
- While viral transfection, using systemic AAV9, is effective in labelling most sensory neuron populations in the dorsal root ganglion, it fails to label C-low threshold mechanoreceptors.

## Introduction

C-fibres are often studied as the peripheral substrates for painful sensory experiences, including their role in chronic pain. However, it has been known for more than 30 years that in humans (Nordin, 1990) and other mammals (Delfini et al., 2013; Kumazawa & Perl, 1977; Leem et al., 1993; Li et al., 2011; Zotterman, 1939) this heterogeneous population of afferents also includes a subset which preferentially responds to low-threshold mechanical stimuli and is thought to drive pleasant touch in humans. This subset of C-fibres, known in humans as C-tactile (CT) fibres and in animals as C-low threshold mechanoreceptors (C-LTMRs) respond to slow and gentle brush stimuli (Bessou et al., 1971; Fang et al., 2005; Löken et al., 2009; Vallbo et al., 1999; Watkins et al., 2017) and low-force skin indentation (Li et al., 2011; Seal et al., 2009; Vallbo et al., 1999; Watkins et al., 2017).

**Evangelia Semizoglou** is a neuroscientist with experience in somatosensory systems and pain biology. She studied biology as an undergrad at Aristotle University of Thessaloniki, in Greece. Then, she received her PhD in neuroscience from King's College London, where she worked with Stuart Bevan, David Andersson, Kim Chisholm and Steve McMahon. Her studies involved pharmacological and physiological investigation of ion channels and opioid receptors. Currently, she is a post-doc at Brigham and Women's Hospital Harvard Medical School under the supervision of William Renthal. Her research revolves around transcriptomic investigation of mouse and human trigeminal ganglia in health and disease. **Dr Kim Chisholm** is an Anne McLaren Fellow at the University of Nottingham, specializing in *in vivo* microscopy and chronic pain research. She completed her PhD at UCL under the supervision of Prof. Kenneth Smith and Prof. Michael Duchen, where she developed innovative techniques to visualize bioenergetic changes in the brain during inflammation and hypoxaemia. Dr Chisholm's postdoctoral work at King's College London, under the late Prof. Stephen McMahon, focused on developing *in vivo* microscopy approaches. Her research now investigates nervous system changes that drive sensitization, aiming to better understand chronic pain mechanisms.

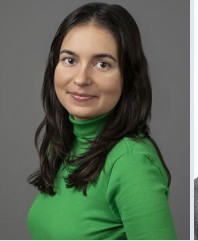
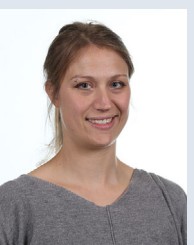

While having first been discovered in cats more than 80 years ago (Zotterman, 1939), CT fibres/C-LTMRs have now been well characterized in humans (Nordin, 1990; Vallbo et al., 1999) and other mammals (Delfini et al., 2013; Kumazawa & Perl, 1977; Leem et al., 1993; Li et al., 2011; Zotterman, 1939). Despite an extended period of study, the methodology used to assess C-LTMR activity and function has remained largely unchanged and technically challenging. Methods including teased fibre recordings in animals and microneurography in humans are labour intensive and very low throughput.

Despite such methods being limited to a small number of specialized groups, our understanding of CT/C-LTMR physiology and its role in the mammalian sensory experience has grown rapidly. We know that both CT fibres in humans and C-LTMRs in many mammals (including non-human primates (Kumazawa & Perl, 1977)), can be found in hairy skin, show slow conduction velocities, respond to low-force indentation of the skin and show vigorous activity in response to slow brushing stimuli at what is considered a pleasant speed, around 1–10 cm/s (Bessou et al., 1971; Fang et al., 2005; Löken et al., 2009; Vallbo et al., 1999; Watkins et al., 2017). Uniquely among mechanoreceptive afferents, increasing stimulus speeds past 10 cm/s reduces CT-fibre activity, resulting in an inverted U-shaped tuning curve to brush speed (Löken et al., 2009). This stimulus–response function is similar in other animals, though shifted to a preference for even slower speeds (Bessou et al., 1971; Watkins, 2022). This stimulus–response function mirrors the reported pleasantness of the brush stimulus, linking CT-fibre activity with perceived pleasantness of a given mechanical stimulus (Ackerley et al., 2014; Löken et al., 2009).

More recently, the study of C-LTMRs has led to a number of discoveries about their gene expression profiles and dependence on ion channels (Middleton et al., 2022; Zheng et al., 2019). Several studies have identified molecular markers of rodent C-LTMRs including vGLUT3 (Larsson & Broman, 2019), Tafa4 (Delfini et al., 2013) and tyrosine hydroxylase (TH) (Li et al., 2011). In particular, TH is a marker of a major group of C-LTMRs and makes up ∼10% of all dorsal root ganglion (DRG) neurons (Li et al., 2011). TH-positive C-LTMRs innervate mouse hairy skin as longitudinal lanceolate endings surrounding hair follicles (Li et al., 2011; Middleton et al., 2022). Li et al. showed, using intracellular recordings, that TH-positive DRG neurons function like human C-LTMRs and have low mechanical thresholds, C-fibre range conduction velocities and respond to cooling stimuli (Li et al., 2011). However, studying C-LTMR excitability at single neuron level is challenging, low throughput and, as a result, often overlooked.

In this paper we discuss a new way to visualize the activity of C-LTMRs, using *in vivo* calcium imaging of the TH-positive C-LTMR population. This technique gives us access to hundreds of DRG neurons simultaneously, facilitating high-throughput assessment of C-LTMRs and their stimulus–response profiles to natural stimuli in a physiologically preserved environment. Using our paradigm, we show that mouse C-LTMRs respond to punctate and dynamic mechanical stimuli, resembling human CT-fibre afferents. Unexpectedly we identify that C-LTMRs are directionally sensitive to brush stimuli.

## Materials and methods

### Ethical approval

All experiments involving mice were performed in accordance with United Kingdom Home Office regulations (*Guidance on the operation of the Animals (Scientific Procedures) Act 1986*) and Laboratory Animal Science Association Guidelines (*Guiding principles on good practice for animal welfare and ethical review bodies*). Experiments were performed in accordance with protocols detailed in a UK Home Office licence.

All mice were group-housed in individually ventilated cages, with free access to food and water, in humidity and temperature-controlled rooms, with a 12 h light–dark cycle.

### Animals

TH$^{\text{CreERT2}}$ (Abraira et al., 2017) mice were sourced from Jackson Labs (Strain #: 025614). To label TH-positive cells transgenically we crossed TH$^{\text{CreERT2}}$ mice with Cre-dependent a tdTomato reporter line (Ai14, Jax, Strain #: 007914) resulting in mice expressing tdTomato in TH-positive neurons (TH-tdTomato). To transgenically express GCaMP in C-LTMRs we crossed the TH$^{\text{CreERT2}}$ mice with the RCL-GCaMP6f mouse line (Ai95D, The Jackson, Laboratory, US, Strain #: 024105) to create a mouse in which GCaMP is expressed in TH-positive cells (TH$^{\text{CreERT2}}$GCaMP6 mouse). Two adult TH-tdTomato mice were used for immunohistochemistry and 10 adult TH$^{\text{CreERT2}}$GCaMP6 mice were used for *in vivo* microscopy.

### Tamoxifen injections

TH-tdTomato and TH$^{\text{CreERT2}}$GCaMP6 mice received tamoxifen in adulthood. Tamoxifen (Sigma) was dissolved at 20 mg/ml in corn oil by sonication for 1 h at 37°C. Once dissolved the tamoxifen was further diluted to 4 mg/ml. Mice received a single I.P. injection at 50 mg/kg and

were used for *in vivo* microscopy or DRG extraction for immunohistochemistry at least 1 week after tamoxifen injection.

### Labelling efficiency of TH-positive cells

**Pup injections.** In two TH-tdTomato mice, sensory neurons were labelled through viral transduction. To achieve this, 5 µl adeno-associated viral vector, serotype 9 (AAV9) expressing GCaMP6s (AAV9.CAG.GCaMP6s.WPRE.SV40, Addgene, USA) was injected subcutaneously in the nape of the neck of pups (P2–P6), using a 10 µl Hamilton syringe. Pups were returned to their home cage. The DRGs were extracted after at least 10 weeks post-injection.

**Immunohistochemistry.** To assess the efficiency of labelling TH-positive cells through systemic injections in pups, we carefully excised the left L4 DRG in adult mice, washed them in PBS and post-fixed them for 2–3 h in 4% paraformaldehyde (PFA). After fixation was complete, DRGs were cryoprotected in 30% sucrose (0.02% sodium azide) overnight. The DRGs were then embedded in optimal cutting temperature compound (Tissue-Tek) to be frozen and cryosectioned into 10 µm sections before mounting onto glass slides.

The tissue was allowed to dry and was preserved at −80°C before further processing. To label cells, the tissue was rehydrated and blocked with 10% goat serum (1 h). The slices were then incubated overnight at 4° with anti-GFP antibodies (Ab13970, Abcam), diluted 1:1000. Following incubation with primary antibody the slides were washed and incubated with goat anti-chicken secondary antibody conjugated to Alexa Fluor 488 (A-11039, Invitrogen), diluted 1:1000, for 2 h at room temperature. DAPI-containing media (Fluoromount-G with DAPI, eBioscience) was used to coverslip the tissue. The DRG sections were imaged with an LSM 710 laser-scanning confocal microscope (Zeiss).

**Evaluation of transfection efficiency.** To assess the transfection efficiency of systemic pup injections of AAV9-GCaMP, 6–10 images were analysed per mouse ($n = 2$). The TH-positive cells were counted by an experienced observer and displayed against the number of cells that were TH- and GCaMP-positive.

### *In vivo* Ca²⁺ imaging of C-LMTRs

At least 1 week after tamoxifen injection, mice were anaesthetized with an initial injection of urethane (12.5% wt/vol) with a dose of 0.3 ml (37.5 mg urethane). After 15 min, additional doses were administered depending on reflex activity until surgical depth was achieved (absence

of hind limb and corneal reflex activity). Mice were placed on a homeothermically controlled heating mat with a rectal probe to maintain their body temperature at 37°C. Their backs were shaved, and an incision was made in the skin over the spinal cord at the level of L3–L5. The muscle and connective tissue overlying the vertebrae were carefully removed and a laminectomy was performed which was extended laterally (to the left side) to include exposure of the L4 DRG. Once the DRG was visible the dura and perineurium were left intact but cleaned with sterile normal saline (0.9%) and cotton buds. The exposed DRG was stabilized between a lateral recumbent and prone position, using spinal clamps (Precision Systems and Instrumentation). This positioning ensured a stabilized DRG focusable using an upright microscope. The DRG was covered with silicone elastomer (World Precision Instruments, Ltd.) to maintain a physiological environment and to prevent it from drying during imaging.

The anaesthetized and stabilized mouse was then placed under Eclipse Ni-E FN upright confocal/multiphoton microscope (Nikon) and the ambient temperature was locally maintained at 32°C and the rectal temperature at 37°C. All time-lapse images were recorded through a 10× air objective at an imaging rate of around 4 Hz with an open pinhole. The open pinhole allowed collection of out-of-focus light and as such reduced the effect of biological movements (Chisholm et al., 2018). A 488 nm Argon ion laser line was used to excite GCaMP and a 500–550 nm filter was used for collection. At the end of the experiment the mouse was killed with an overdose of anaesthetic while imaging was continued for a minimum of 30 min post-mortem to allow accumulation of calcium inside cell bodies.

**Brush stimulation.** During *in vivo* microscopy the mouse leg was stimulated with a rotating brush stimulator. All mice undergoing *in vivo* microscopy were stimulated with the rotating brush ($n = 10$). To create a standardized brush stimulus, the head of a make-up brush with wide soft bristles (to maximise the stimulation area while minimizing the impact of brush distance: Boots, Lime, Crime Aquarium Brush Set) was attached to a motor (RS Components, RS PRO Brushed Geared motor). The motor was driven by a variable current power source (Hanmatek HM305 bench supply with constant current/voltage output). The relationship between voltage input and brush speed is described in Table 1. The direction of the brush could be reversed by alternating polarity. The speed of 1.55 cm/s was the slowest speed reliably achieved. Given that the natural growth orientation of the leg hair in mice is to grow from proximal to distal direction, so that the tip of the hair is pointing towards the paw, the brush was placed such that it would brush from torso to paw (with

**Table 1. Relationship between voltage input and brush speed.**

| Speed | Voltage |
|---|---|
| 1.55 cm/s | 1.5 V |
| 3 cm/s | 2.9 V |
| 6 cm/s | 5.8 V |
| 10 cm/s | 9.6 V |
| 20 cm/s | 19.2 V |
| 30 cm/s | 29 V |

the hair) and in the reverse direction (against the hair). The brush was allowed to rotate four times at every speed before the speed was changed in a random order.

**Von Frey stimuli.** To assess responses to punctate stimuli, von Frey filaments with difference bending forces were used to stimulate across four locations on the leg each, for a total of 12 stimuli across 12 sites, covering the same stimulation area as the brush stimulus. Nine mice received von Frey stimulation during *in vivo* microscopy. The following filaments were applied in random order: 0.07 g, 0.16 g and 0.4 g.

**Time lapse analysis.** Time-lapse recordings were aligned to a reference frame using the Nikon Imaging Software Align Application (Elements AR0.30.01). The resulting stabilized recording was used to extract fluorescence traces over time using Fiji/ImageJ version 2.3.0. The free-hand selection tool was used to select regions of interest (ROIs) for each cell. To select all cell bodies, including those from neurons that did not respond to external stimuli, mice were killed under the microscope and the calcium signal was allowed to increase in the cells. An additional ROI for the background was selected over a region with no cell bodies present. The fluorescence intensity of the pixels present in each ROI was averaged for each time frame, resulting in a single fluorescence trace per ROI. The resulting traces were processed further using R version 3.6.1 and RStudio version 1.2.5001. The background trace was subtracted from each cell body ROI at each time frame. The resulting background-subtracted traces were normalized using the following formula:

$$\frac{\Delta F}{F} = \frac{F_t - F_0}{F_0},\qquad(1)$$

where $F_t$ is the fluorescence intensity at time $t$ and $F_0$ is the averaged fluorescence intensity over a baseline period, prior to the onset of any stimulation (here a 20 s period starting 40 s prior to the first stimulus).

To define responsiveness over a stimulation period an objective threshold was set which would categorize a cell as responding. The objective threshold was set

by comparing its performance against a gold standard (manual selection of responding cells by an experienced observer) on a subset of the data. Of the thresholds tested, the following was found to be the most accurate: a response is present if:

$$F_{\max} > 1.1\,(\max F_{Bl}) + 5\,(\sigma\,F_{Bl}),\qquad(2)$$

where $F_{\max}$ is the maximal fluorescence intensity over the stimulation period, $\max F_{Bl}$ is the maximum fluorescence intensity over a baseline period (defined relative to each stimulation period, here defined as 5 s starting 10 s before the stimulus) and $\sigma\,F_{Bl}$ is the standard deviation over said baseline period. With the chosen parameters, the experienced observer (gold standard) and the objective threshold agree on approximately 95.5% of the test cases.

### Statistical analysis

Data are presented as means ± standard deviation (SD). Analysis and graphing were conducted using Wolfram Mathematica (version 12.1,) R (version 3.6.1) and R studio (version 1.2.5001) and Microsoft Office Excel (version 2108).

Repeated measure ANOVAs (with corrected degrees of freedom for sphericity violations, as reported in the text) were used to assess the effect of brush stimulation speed and direction, the effect of repeated brush stimulation and the effect of von Frey stimulation on peak calcium intensity and area under the curve (AUC). Bonferroni-corrected pairwise comparisons were used to assess the effect of brush direction at discreet speeds and to assess the difference in probability of a response to punctate/brush stimuli, given a response to the other stimulation modality (i.e. P(punctate | brush) and P(brush | punctate).

**Analysis for repeated brush stimuli.** Analysis for the run-off of responses to brush stimuli was performed on responses to the slowest brushing speed (1.5 cm/s) in Wolfram Mathematica (v12.1). Due to slight variations in the onset of each brush stimulus, we had to determine the exact timing of the brush responses for each animal. To do this, the average trace of responding cells (response defined as in 'Time lapse analysis') was determined per mouse. From this average trace the location of four 'brush windows' was determined for each mouse (equivalent to the four peaks of neuronal responses to four brush stimuli) using the 'FindPeaks' function. To optimise peak detection the following parameters were modified: 'blurring scale' ($\sigma$), 'sharpness' ($s$) and 'minimum peak value' ($t$), until four peaks were identified in the average trace of each animal ('average peaks' for brevity).

The time window of the four average peaks ± 5 s was considered the 'brush window' of each brush stimulus.

Four such windows were determined per mouse per brush direction (four windows 'with the hair' and four windows 'against the hair'). These windows were used to select the maximum response to each brush stimulus per cell (four peak responses 'with the hair' and four peak responses 'against the hair' direction).

Maximum responses/peaks were averaged across cells, per animal. Each animal provided one *N*. Data from 'with the hair' were excluded from one mouse as no responding cells could be identified. Data from 'against the hair' were excluded from a separate mouse as four peaks could not reliably be identified. With excluded data we had *N* = 9 for each direction.

## Results

### Post-natal injections of AAV9-GCaMP poorly labelled TH-positive sensory neurons

To establish a protocol for labelling TH-positive C-LTMRs with the genetically encoded calcium indicator GCaMP6 we made use of existing labelling protocols for primary afferent neurons (Wang et al., 2018). As previously described (Wang et al., 2018), new-born TH-tdTomato pups (2–6 days post-natal) received subcutaneous injections of adeno-associated virus, subtype 9 (AAV9) expressing GCaMP. Mice were perfused between 8 and 10 weeks later, and the labelling efficiency of GCaMP in TH-positive cells was assessed (Fig. 1).

This approach did not label sufficient C-LTMR neurons with GCaMP when assessed in adulthood (Fig. 1*A*,*B*): among 118 TH-tdTomato positive cells observed, only eight were found to also express GCaMP6 (Fig. 1*B*). To improve the number of GCaMP+ C-LTMRs, we took an alternative approach and crossed the TH$^{CreERT2}$ line with the flox-STOP-*GCaMP6f* line to generate TH$^{CreERT2}$GCaMP6 mice, which, following tamoxifen administration, expressed GCaMP selectively in TH-positive neurons/C-LTMRs. The labelling efficiency was sufficient to visualize a strong GCaMP signal *in vivo*, after exposure and stabilization of the L4 DRG (Fig. 2).

To visualize C-LTMR activity *in vivo* (Fig. 2*A*) we exposed the L4 DRG in an anaesthetized mouse and visualized the calcium signals using standard single-photon microscopy (confocal microscope with an open pinhole to reduce the effects of tissue curvature and biological movement, as described in Chisholm et al. (2018)). An automated rotating brush stimulus (Fig. 2*B*) was used to reproducibly stimulate the ipsilateral leg with low-threshold brush stimulation, as shown in Fig. 2. The brush was constructed from a soft make-up brush head (due to its wide and soft bristles), attached to a rotating motor, the speed of which was controlled by a variable current power source. Changes in voltage input altered the brush speed in a direct way (Table 1). The direction of the brush could be reversed by alternating polarity. Von Frey stimuli were applied to the same area as the brush stimulus (Fig. 2).

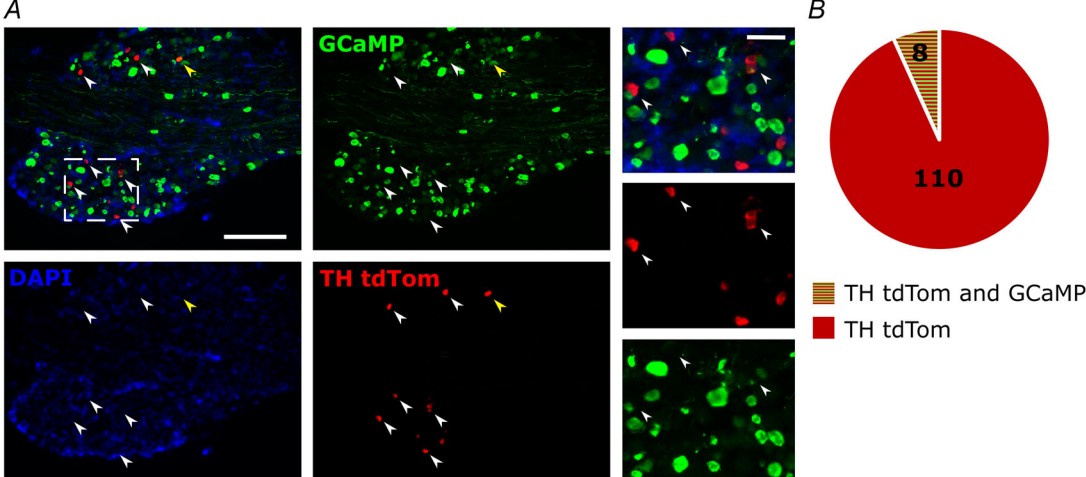

**Figure 1. Labelling efficiency of systemic AAV9-GCaMP in TH$^{CreERT2}$tdTomato mice**

*A*, representative images of TH and GCaMP-positive cells after systemic injections of AAV9-GCaMP in TH-tdTomato pups. Immunohistochemistry was performed on DRGs excised from adult mice: TH (red), GCaMP (green) and DAPI (blue). White arrow heads indicate examples of TH-tdTomato positive cells, yellow arrow heads indicated examples of double positivity (TH-tdTomato and GCaMP positivity). Left overview images of whole DRG sections, scale bar = 200 μm. Close up inserts (dotted white line) shown on the right, scale bar = 50 μm. *B*, quantification of *A*: overlap between TH-tdTomato positivity and GCaMP positivity. Proportion of TH-tdTomato positive cells is indicated in red (*n* = two animals, 110 positive cells), double positivity (TH-tdTomato and GCaMP-positive cells) is indicated in red and green stripes (*n* = two animals, eight double positive cells). [Colour figure can be viewed at wileyonlinelibrary.com]

## TH-positive C-LTMRs respond to brush stimuli in a graded, directionally specific way

To understand the response profile of mouse C-LTMRs to brush stimuli, a motor-controlled brush stimulus was applied to the ipsilateral leg, and it was sufficient to consistently and reproducibly activate L4 afferents expressing TH (Fig. 3). Brushing with the grain of the hair revealed a very strong speed-dependent activity pattern in TH-positive neurons, in which slower stimuli preferentially activated TH-positive neurons compared with faster stimuli (Fig. 3*A*–*C*), up to 10 cm/s. When the speed is increased to more than 10 cm/s of brushing speed, the relationship between stimulus speed and response intensity is less pronounced, plateauing when brushing with the hair while showing a small peak at 20 cm/s when brushing against the hair (Fig. 3*B*–*C* and Movie 1).

Interestingly, when brushing against the hair the responses of TH-positive cells were enhanced compared to brushing with the hair, at every speed (Fig. 3*A*–*D* and Movie 1), with a particularly strong discrepancy at 20 cm/s (Fig. 3*C, D*). As Mauchly's test indicates a violation of sphericity in all cases, the degrees of freedom were corrected using Greenhouse–Geisser estimates of sphericity. There was a significant effect of both speed and direction, both when analysed by max fluorescence intensity (Fig. 3*C*, speed: $F_{(2,16)} = 65.632$, $P < 0.001$, effect size estimate: $\eta^2_p = 0.89$; direction, $F_{(1,8)} = 46.4$, $P < 0.001$, effect size estimate: $\eta^2_p = 0.85$) and AUC (Fig. 3*D*, speed: $F_{(1,9)} = 116.221$, $P < 0.001$, effect size estimate: $\eta^2_p = 0.936$; and direction, $F_{(1,8)} = 59.832$, $P < 0.001$, effect size estimate: $\eta^2_p = 0.882$), with an interaction between direction and speed only present when analysis was performed by AUC ($F_{(1,10)} = 15.289$, $P = 0.002$, effect size estimate: $\eta^2_p = 0.656$). It should be noted that the effect of brush speed is more pronounced in the AUC analysis, likely in part due to the width of the response decreasing when the speed and frequency of the stimulus increase. Bonferroni-corrected pairwise comparisons between brushing were conducted and response intensities were significantly reduced when brushing with the hair compared to brushing against the hair at every speed measured and with both analytical approaches (max intensity, Fig 3*C*, and AUC, Fig 3*D*). For max intensity at 1.55 cm/s: $t_{(8)} = 3.854$, $P = 0.05$; at 3 cm/s: $t_{(8)} = 4.008$, $P = 0.004$; at 6 cm/s: $t_{(8)} = 6.273$, $P < 0.001$; at 10 cm/s: $t_{(8)} = 7.857$, $P < 0.001$; at 20 cm/s: $t_{(8)} = 5.014$, $P = 0.001$ and at 30 cm/s: $t_{(8)} = 5.126$, $P < 0.001$;

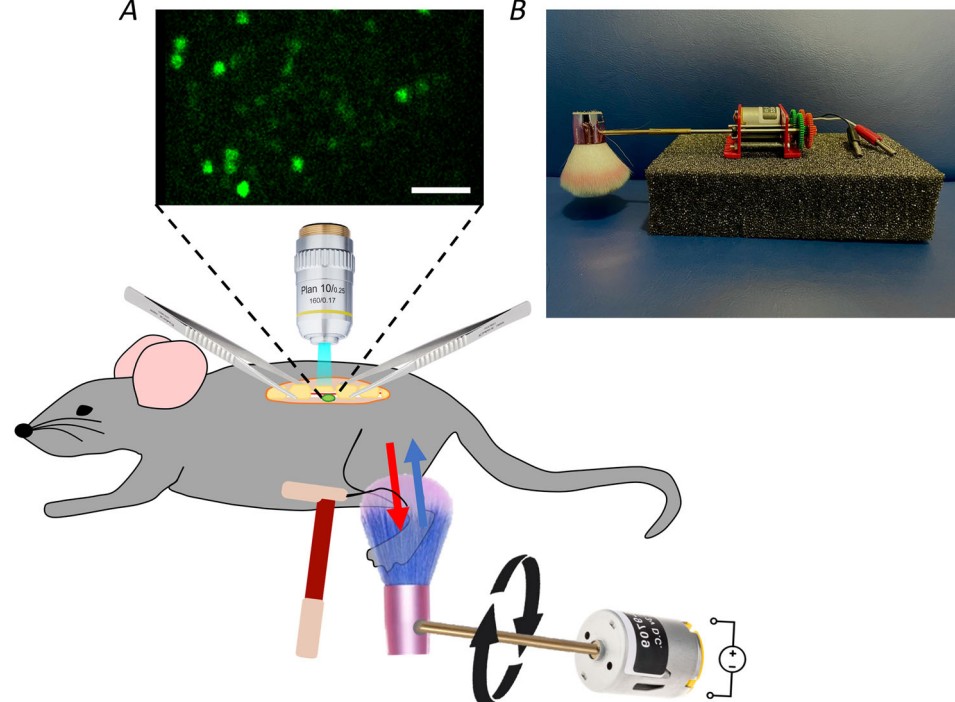

**Figure 2. *In vivo* microscopy set-up for DRG imaging**
*A*, set-up for *in vivo* microscopy of L4 DRG. After anaesthesia the L4 DRG is exposed and stabilized through clamps. Standard single-photon microscopy is used to visualize neurons in the DRG. During imaging peripheral stimuli are applied to the leg of the mouse. To ensure controlled and reproducible brush stimulation a brush head was attached to a rotating motor, which provided control over the speed and direction of the brush stimulus. Red arrow indicates brushing with the hair, blue arrow indicates brushing against the hair. Punctate mechanical stimuli are applied to the same region using von Frey filaments. *B*, picture of custom-made rotating brush stimulus. [Colour figure can be viewed at wileyonlinelibrary.com]

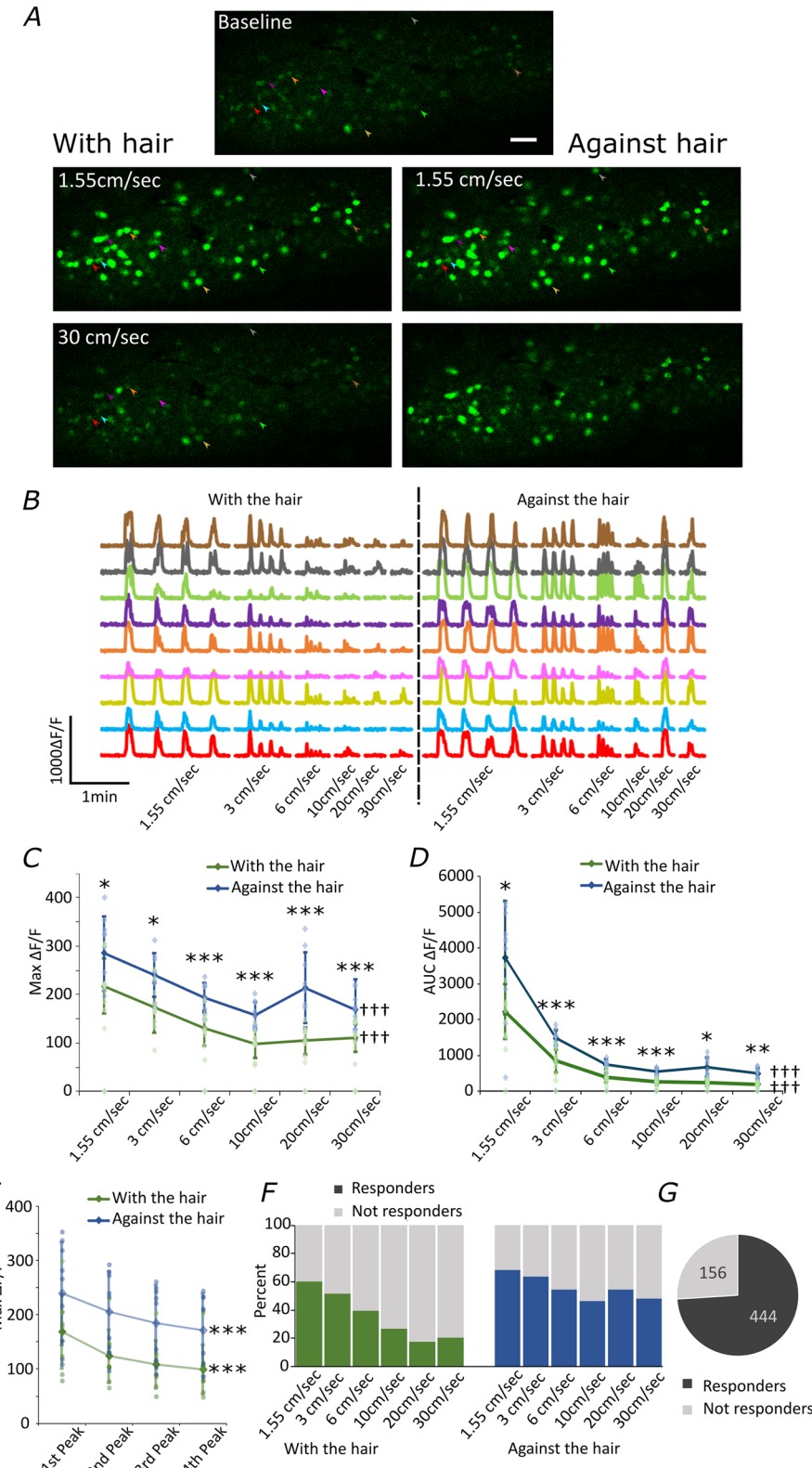

**Figure 3. Speed and directional selectivity of TH-positive cells**
*A*, representative images of calcium signal from DRGs recorded *in vivo* during brush stimulation of the ipsilateral leg. Scale bar = 100 μm. *B*, representative traces of TH-positive primary afferents responding to brush stimulation of the ipsilateral leg at different speeds. Each speed was repeated four times. Quantification of *C*, maximum calcium signal intensity and *D*, area under the curve (AUC) in cells responding to different speeds of brush. Only positive responses (as defined in *Materials and methods*) were included. *N* = 10 mice (total recorded cells = 600).

Maximum calcium signal intensity and AUC were significantly higher when brushing against the hair at every speed, compared with brushing with the hair: * $P \leq 0.05$, ** $P \leq 0.005$, *** $P \leq 0.001$ (for full statistical description see text). There was a main effect of speed and direction for both max intensity and AUC (see text for details): ††† $P \leq 0.001$. Data displayed as means ± SD. *E*, quantification of the run-off of neuronal responses during repeated stimulation. Graph showing the maximum intensity (peaks) of neurons during four cycles of brush stimulation at 1.5 cm/s. Four peaks (one at each brush interval) were identified for each cell and averaged per animal. $N = 9$ (one animal data set was removed in the forward direction and one in the reverse direction because peaks could not be reliably detected). There was a significant run-off of responses from first to fourth peak as determined by a repeated measures ANOVA: *** $P \leq 0.001$ (see text for details). Data displayed as means ± SD. *F*, quantification of the percentage of cells responding to different speeds of brush. $N = 10$ mice (total recorded cells = 600). *G*, proportion of cells responding to any brush stimulus. $N = 10$ mice (total recorded cells = 600). See also Movie 1. [Colour figure can be viewed at wileyonlinelibrary.com]

Fig. 3*C*). For AUC at 1.55 cm/s: $t(8) = 4.990$, $P = 0.006$; at 3 cm/s: $t(8) = 6.824$, $P = 0.001$; at 6 cm/s: $t(8) = 6.726$, $P = 0.001$; at 10 cm/s: $t(8) = 6.456$, $P = 0.001$; at 20 cm/s: $t(8) = 5.076$, $P = 0.006$ and at 30 cm/s: $t(8) = 6.048$, $P < 0.002$; Fig. 3*D*.

We also observed a reduction in responsiveness (run down) after repeated rounds of brushing at the very slow speed (Fig. 3*E*). Faster speeds were not assessed because peak detection was most reliable at 1.5 cm/s. The peak intensity of the calcium transients declined from first to fourth brush (each brush was repeated four times) both when brushing with the direction of the hair and brushing against the direction of the hair. As Mauchly's test indicated a violation of sphericity the degrees of freedom were corrected with Greenhouse–Geisser estimates of sphericity. There was a significant decrease in the peak calcium intensity from first to fourth brush stimulus, both in the 'with the hair' $F(1.4,11.2) = 43.6$, $P < 0.001$ and 'against the hair' direction: $F(1.3, 10.5) = 29.6$, $P < 0.001$. The effect size estimates were $\eta^2_p = 0.85$ and $\eta^2_p = 0.79$, respectively.

A similar relationship was observed in the percentage of cells recruited by each stimulus, with more cells recruited with slower than with faster speeds and an overall increased percentage of cells recruited when brushing against the hair, compared with brushing with the hair (Fig. 3*F*).

Overall, we were able to record at least a single response from 74% of recorded cells (Fig. 3*G*), suggesting good recruitment of TH-positive cells in the L4 DRG through brush stimulation of the leg. The total number of cells labelled with GCaMP was detected post-mortem when calcium signals increased inside the bodies of cells, revealing all labelled cells.

### TH-positive cells respond to punctate stimuli

The response of C-LTMRs to punctate stimuli is still under debate. To address this question, we used von Frey filaments of different strengths applied at different locations across the leg (covering the same stimulation area as the brush stimulus).

We were able to activate TH-positive cells with punctate stimuli ranging from 0.07 to 0.4 g (Fig. 4 andMovie 2). While the activation level of TH-positive cells (as measured by peak GCaMP signal) remained stable across stimulation intensities (Fig. 4*D; F* (2,16) = 2.997, $P = 0.078$), higher percentage of responding cells were recruited at higher forces (Fig. 4*E; F* (2,16) = 28.053, $P < 0.001$).

The von Frey stimuli were applied over the same area of the leg as the brush stimuli but covered a reduced area. This reduced area of stimulation (due to the very discrete nature of von Frey stimulation) likely contributes to the relatively low recruitment rate of TH-positive neurons with von Frey stimuli (just under 50%, Fig. 4*F*). However, as all receptive fields stimulated with von Frey stimuli were also stimulated by the brush stimulus we were able to determine the percentage of cells responding to von Frey stimuli which also responded to brush stimulation, as a measure of specificity to different types of mechanical stimuli (i.e. brush *vs.* punctate stimuli) (Fig. 4*G*). We found that the majority of TH-positive neurons responding to punctate mechanical stimuli also responded to brush stimuli (95%). Additionally, there was no difference in the probability of one type of mechanical stimulation, given another. For example, there was no difference in the probability of a response to punctate stimulation (von Frey at 0.07 g, 0.16 g and/or 0.4 g) in cells responding to brushing with the hair *vs.* against the hair $t(8) = 1.357$, $P = 0.106$ (Fig. 4*H*) and there was no difference in the probability of any brush response (with the hair and/or against the hair) given a response to punctate stimulation at 0.07 g *vs.* 0.16 g *vs.* 0.4 g: $F(2, 16) = 0.22$, $P = 0.805$ (Fig. 4*I*).

Collectively, we have used targeted genetics combined with *in vivo* calcium imagining to characterize in detail the mechanosensitivity of C-LTMRs innervating the hind leg.

## Discussion

In this paper we describe a new approach for the assessment of C-LTMR function by measuring $Ca^{2+}$ flux as a proxy for neuronal activity. We focused on characterizing the mechanosensitivity of these afferents

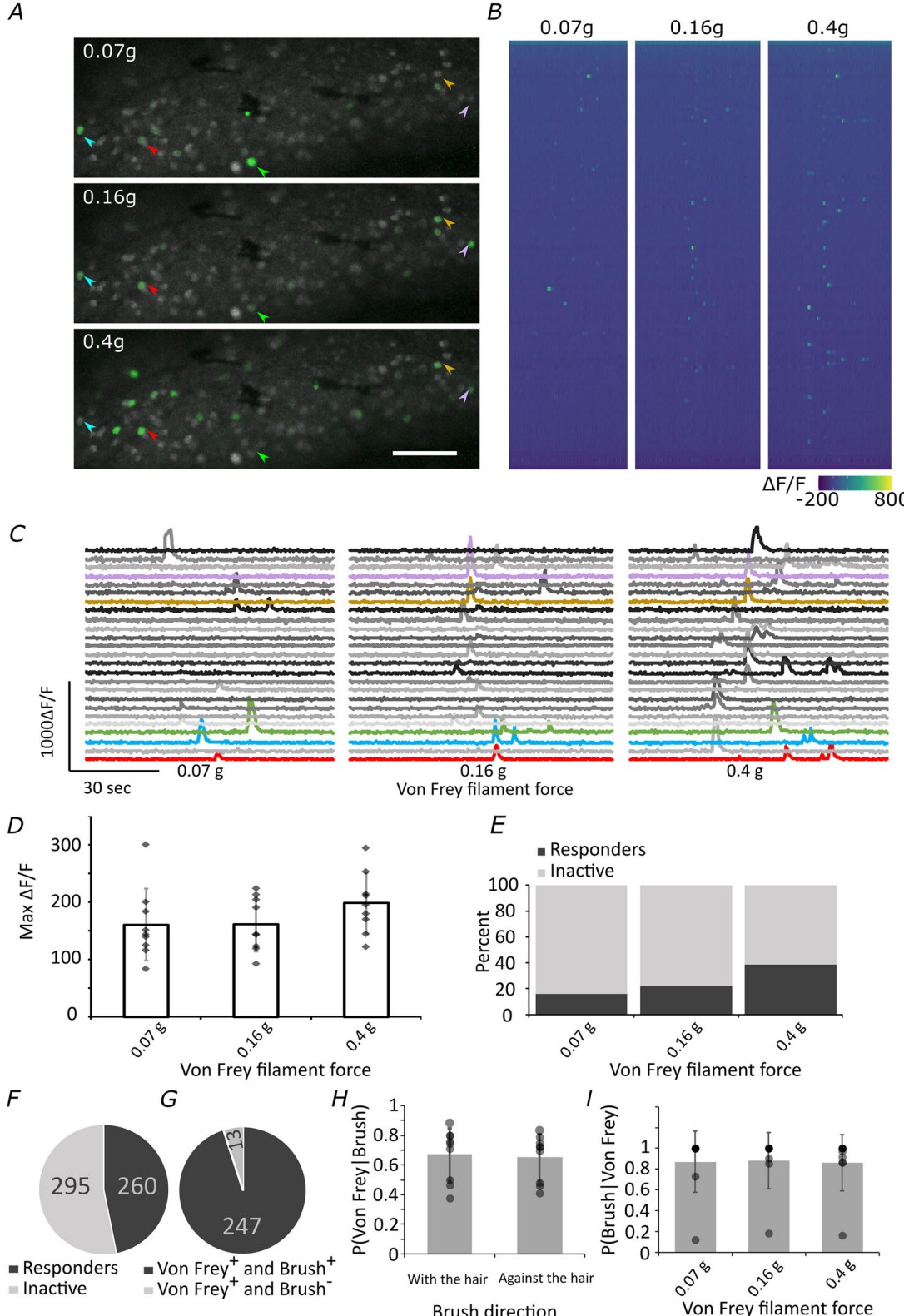

**Figure 4. Responses of TH-positive cells to punctate von Frey stimuli**
*A*, representative images of calcium signal from DRGs recorded *in vivo* during von Frey stimulation of the ipsilateral leg. In green: $\Delta F/F$ of standard deviation signal across application of four separate von Frey stimuli on the leg. In grey: standard deviation signal across baseline, to visualize cells that were GCaMP-positive but not activated by von Frey stimulation. Arrows indicate cell bodies that have been activated by von Frey stimulation. Scale bar

= 100 µm. *B*, heat map of responses in a single mouse DRG during von Frey stimulation. *N* = 115 cells. *C*, representative traces of TH-positive primary afferents responding to von Frey stimulation of the ipsilateral leg at different forces. Each weight was repeated four times at different positions across the leg. Coloured cells represent the same cells as highlighted in *A* with arrows. *D*, quantification of calcium signal intensity in cells responding to von Frey stimulation of different filament forces. Only cells responding were included and the maximum intensity of the response was averaged across cells. *N* = 9 (total recorded cells = 555). Data displayed as means ± SD. *E*, percentage of TH-positive cells responding at different von Frey filament force strengths. *N* = 9 (total recorded cells = 555). *F*, number of cells responding at least once to a von Frey stimulus. *N* = 9 (total recorded cells = 555). *G*, number of cells responding to von Frey only (von Frey$^+$ Brush$^-$) *vs.* cells responding to von Frey and Brush stimuli (von Frey$^+$ Brush$^+$). *N* = 9 animals (total number of cells responding to von Frey (with or without brush) = 260 cells). See also Movie 2. *H*, to assess the relationship between responses to punctate and brush stimuli we assessed the conditional probability of a response to any punctate mechanical stimuli (von Frey stimulation of 0.07 g, 0.16 g and/or 0.4 g) given a response to brushing with the hair *vs.* against the hair. Data displayed as means ± SD. *N* = 9 animals. Equally, *I*, conditional probability of a response to a brush stimulus (with the hair and/or against the hair), given a response to punctate mechanical stimulation with von Frey at 0.07 g *vs.* 0.16 g *vs.* 0.4 g. Data displayed as means ± SD. *N* = nine animals. [Colour figure can be viewed at wileyonlinelibrary.com]

*in vivo* and believe this approach has the potential to complement more traditional, electrophysiological approaches, providing large-scale data on the activity of multiple C-LTMR sensory neurons simultaneously. This study has been facilitated through the genetic identification of C-LTMRs using the TH$^{CreERT2}$ driver mouse line (Li et al., 2011) providing selectively directed GCaMP expression to C-LTMRs. This has allowed us to functionally assess the mechanosensitivity of this specialized sensory neuron subpopulation to punctate and dynamic touch *in vivo*.

A common approach of labelling primary afferents is through the systemic or local injection of AAV particles containing transgenes, such as GCaMP. In particular, intraplantar and systemic AAV administration in neonatal mice has been used successfully to target sensory neurons (Ingram et al., 2023; Wang et al., 2018). Our initial strategy was to transduce all sensory neurons using AAV9-GCaMP in mice where C-LTMRs were genetically labelled with a red florescent protein. This method, however, was inefficient and unsuccessful in labelling TH-positive C-LTMRs. While the reason remains unclear, another study failed to see C-LTMR infection following intraneural AAV9 and speculated that C-LTMRs may be partially resistant to this serotype (Bernal Sierra et al., 2017).

To circumvent this issue, we crossed TH$^{CreERT2}$ mice with the Cre-dependent RCL-GCaMP6 mouse line, which resulted in GCaMP expression only in TH-positive neurons, following tamoxifen injection. Using this approach, we were successfully able to visualize calcium transients in C-LTMRs in anaesthetized mice. We combined *in vivo* calcium imaging of the exposed DRG with natural stimuli applied to the hairy skin of the ipsilateral leg. These stimuli included automated brush stimulation and low force punctate stimuli. Both stimuli were able to effectively activate C-LTMR neurons in mice and resulted in easily identifiable calcium transients in their cell bodies.

Using this technique, we report sensitivity to low-intensity punctate stimuli with responses to von Frey stimuli as low as 0.07 mN, consistent with the high sensitivity to punctate stimuli reported in rats (Leem et al., 1993; Seal et al., 2009), mice (Li et al., 2011; Middleton et al., 2022) and humans (Vallbo et al., 1999; Watkins et al., 2017).

Intravital microscopy combined with repeated brush stimuli at slow speeds also showed a familiar run down of responses to repeated stimuli. This fatigue is a characteristic feature of C-LTMRs and has been extensively described in humans and experimental animals (Andrew, 2010; Bessou et al., 1971; Iggo & Kornhuber, 1977; Nordin, 1990). In addition, *in vivo* calcium imaging was able to reveal that C-LTMR activity shows the same preference to slower brushing speeds as observed in CT fibres/C-LTMRS in human (Löken et al., 2009; Nordin, 1990; Vallbo et al., 1999) and other mammals (Delfini et al., 2013; Kumazawa & Perl, 1977; Shea & Perl, 1985; Zotterman, 1939). However, due to technical constraints, we were unable to stimulate at less than 0.5 cm/s – as the motorized brush stimulus was no longer able to brush at a continuous speed below 0.5 cm/s.

This unique relationship between brushing speed and neuronal responses in C-LTMRs has strongly implicated them in the encoding of pleasant touch. Indeed, more than 10 years ago Löken and colleagues found that the inverted U-shaped response curve of CT fibres to increasing brush speed (with preferred brushing velocities around 1–10 cm/s) was strongly correlated with the perceived pleasantness of the stimulus (Löken et al., 2009). Building on this finding, responses of CT fibres to stroking seem to be modulated by temperature. Brush stimuli at neutral temperatures which are typical of skin temperature, result in preferential activation of CT fibres (Ackerley et al., 2014). These findings, together with patient-derived data (suggesting that C-fibres are necessary (Morrison et al., 2011) and sufficient (Cole et al., 2006; Olausson et al., 2002, 2008) for the detection of pleasant, light touch), were instrumental in associating CT-fibre activity with

dimensions of affective and social touch (Liljencrantz & Olausson, 2014; McGlone et al., 2014; Morrison et al., 2010; Olausson et al., 2010).

Animal studies similarly reflect the link between C-LTMR activity and positive affect. Stimulation of C-LTMRs resulted in conditioned place preference (CCP), indicative of reward (Huzard et al., 2022). For example, an intersectional viral chemogenetic approach to stimulate neurons positive for Nav1.8 and $Ca_V3.2$ (an ion channel expressed by and required for normal function of C-LTMRs) was rewarding (as assessed through the CCP paradigm) and promoted touch-seeking behaviours (Huzard et al., 2022). Conversely, functional deficiency of C-LTMRs (by gene ablation of $Ca_V3.2$) induced social isolation and reduced tactile interactions (Huzard et al., 2022).

Despite this impressive repertoire of findings, associating CT fibres/C-LTMRs with pleasant/affective touch, it remains difficult to causally link CT fibres/C-LTMR activity to the perception of pleasant touch. In healthy humans the activation of CT fibres necessitates the activation of A$\beta$ fibres, precluding isolated stimulation; and while it is possible to selectively modulate C-LTMR activity in animals, the assessment of the pleasantness of a stimulus is not straightforward.

Using *in vivo* calcium imaging we were able to show that TH-positive cells respond more strongly to brushing stimuli which are applied against the hair as compared with brushing with the hair. Interestingly, in pets with fur, petting/brushing in the direction of hair growth is recommended to avoid stress and discomfort (Casalia et al., 2017; Herron & Shreyer, 2014), suggesting that brushing against the direction of the hairs is less pleasant. Assuming that such a relationship between grooming direction and pleasantness is seen in our experimental mice, the reported increase in responsiveness of C-LTMRs when brushing against the hair could provide a rare case of dissociation between the pleasantness of a stimulus (against the hair being less pleasant than with the hair) and the activity of TH-positive C-LTMRs (more active when brushing against the hair, compared with brushing with the hair). Indeed, it is likely that to perceive the complex sensory experience of affective touch, direct and unmodulated signalling from a single afferent population is not sufficient. Instead, a combination of sensory signals (peripheral and central) will create the rich experience that is pleasant touch.

There are a number of possible reasons for the observed directional selectivity of C-LTMR responses, including: (a) a wider excursion of the hair when brushed against its natural orientation, which may cause more prolonged and effective activation of C-LTMRs, (b) a possible differential expression of mechanosensitive channels around the hair cell, or (c) the variation in tension applied to potential protein tethers synthesized by sensory neurons (Hu et al., 2010). It should be noted that murine A$\delta$-LTMRs innervating hairy skin have also been shown to be preferentially tuned to deflection of body hairs in the caudal-to-rostral direction (Rutlin et al., 2014) (which would broadly equate to our categorization of deflection 'against the hair'). This tuning property is thought to be explained by the finding that A$\delta$-LTMR lanceolate endings around hair follicles are polarized and concentrated on the caudal side of each hair follicle (Rutlin et al., 2014). Similarly, A$\delta$-LTMR directional sensitivity has been shown in rodent glabrous skin, although these afferents appear to be preferentially tuned in the oppositive direction (similar to our 'with the hair' category), while lanceolate endings are polarized in the same way as hairy skin (Walcher et al., 2018). This suggests that lanceolate ending polarization may not fully explain LTMR directional sensitivity and other mechanism are likely at play. Indeed, C-LTMR lanceolate endings do not show hair follicle polarization.

## Conclusion

We were able to show that calcium imaging in DRG neurons is a suitable technique to visualize the activity of C-LTMR neurons in anaesthetized mice. We were able to show expected responses to mechanical stimuli, including activation by low-threshold punctate stimuli and preferential responses for slow brushing speeds. We also show an unexpected directional sensitivity of C-LTMRs for brushing against the hair rather than with the hair.

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

## Additional information

### Data availability statement

The data and analysis pipelines that support the findings of this study can be made available from the corresponding author (K.I.C.), upon request.

### Competing interests

The authors declare that they have no conflicts of interest.

### Author contributions

K.I.C., D.L.B. and S. J. M. conceived the project and designed the experiments. E.S. and L.L.R. collected the data. K.I.C. and T.T. analysed the data. E.S., L.L.R., S.J.M., J.P.S., T.T., D.L.B. and K.I.C. wrote and edited the manuscript.

### Funding

This work was supported by an Anne McLaren Fellowship, University of Nottingham, held by K.I.C. and funding from the UK Medical Research Council (grant ref. MR/T020113/1), held by D.L.B.

### Keywords

brush, calcium imaging, C low threshold mechanoreceptors, GCaMP, hair, *in vivo* microscopy, mechanical stimuli, mechanosensation

### Supporting information

Additional supporting information can be found online in the Supporting Information section at the end of the HTML view of the article. Supporting information files available:

**Peer Review History**
**Movie 1**
**Movie 2**

