## [Peer Review History · The Journal of Physiology]

In vivo calcium imaging reveals directional sensitivity of C-low threshold mechanoreceptors

Evangelia Semizoglou, Laure Lo Re, Steven J Middleton, Jimena Perez-Sanchez, Tommaso Tufarelli, David L Bennett, and Kim I Chisholm

DOI: 10.1113/JP286631

Corresponding author(s): Kim Chisholm (kim.chisholm@nottingham.ac.uk)

Review Timeline:

Submission Date:	28-Mar-2024
Editorial Decision:	21-May-2024
Revision Received:	18-Sep-2024
Editorial Decision:	30-Oct-2024
Revision Received:	13-Nov-2024
Accepted:	28-Nov-2024

Senior Editor: Katalin Toth

Reviewing Editor: Nathan Schoppa

Transaction Report:

Dear Dr Chisholm,

Re: JP-RP-2024-286631 "In vivo calcium imaging reveals directional sensitivity of C-low threshold mechanoreceptors" by Evangelia Semizoglou, Laure Lo-Re, Steven J Middleton, Jimena Perez Sanchez, Tommaso Tufarelli, David L Bennett, and Kim I Chisholm

Thank you for submitting your manuscript to The Journal of Physiology. It has been assessed by a Reviewing Editor and by 2 expert referees and we are pleased to tell you that it is potentially acceptable for publication following satisfactory major revision.

LANGUAGE EDITING AND SUPPORT FOR PUBLICATION: If you would like help with English language editing, or other article preparation support, Wiley Editing Services offers expert help, including English Language Editing, as well as translation, manuscript formatting, and figure formatting at www.wileyauthors.com/eoo/preparation. You can also find resources for Preparing Your Article for general guidance about writing and preparing your manuscript at www.wileyauthors.com/eoo/prepresources.

REVISION CHECKLIST:

Please upload two versions of your manuscript text: one with all relevant changes highlighted and one clean version with no changes tracked. The manuscript file should include all tables and figure legends, but each figure/graph should be uploaded as separate, high-resolution files. The journal is now integrated with Wiley's Image Checking service. For further details, see: <https://www.wiley.com/en-us/network/publishing/research-publishing/trending-stories/upholding-image-integrity-wileys->

image-screening-service

We look forward to receiving your revised submission.

Yours sincerely,

Katalin Toth
Senior Editor
The Journal of Physiology

REQUIRED ITEMS

- Include a Key Points list in the article itself, before the Abstract.
- Author photo and profile. First or joint first authors are asked to provide a short biography (no more than 100 words for one author or 150 words in total for joint first authors) and a portrait photograph. These should be uploaded and clearly labelled together in a Word document with the revised version of the manuscript. See Information for Authors for further details.
- You must start the Methods section with a paragraph headed Ethical Approval. A detailed explanation of journal policy and regulations on animal experimentation is given in Principles and standards for reporting animal experiments in The Journal of Physiology and Experimental Physiology by David Grundy J Physiol, 593: 2547-2549. doi:10.1113/JP270818). A checklist outlining these requirements and detailing the information that must be provided in the paper can be found at: <https://physoc.onlinelibrary.wiley.com/hub/animal-experiments>. Authors should confirm in their Methods section that their experiments were carried out according to the guidelines laid down by their institution's animal welfare committee, and conform to the principles and regulations as described in the Editorial by Grundy (2015), including an ethics approval reference number. The Methods section must contain a statement about access to food, water and housing, details of the anaesthetic regime: anaesthetic used, dose and route of administration, and method of killing the experimental animals.
- The reference list must be in alphabetical order, rather than numbered, to comply with our Journal format.
- Your manuscript must include a complete Additional Information section, including competing interests; funding; author contributions and acknowledgements.
- The Journal of Physiology funds authors of provisionally accepted papers to use the premium BioRender site to create high resolution schematic figures. Follow this link and enter your details and the manuscript number to create and download figures. Upload these as the figure files for your revised submission. If you choose not to take up this offer, we require figures to be of similar quality and resolution. If you are opting out of this service to authors, state this in the Comments section on the Detailed Information page of the submission form. The link provided should only be used for the purposes of this submission. Authors will be charged for figures created on this premium BioRender account if they are not related to this manuscript submission.
- Please upload separate high-quality figure files via the submission form.

- Please ensure that the Article File you upload is a Word file.

- Please include an Abstract Figure file, as well as the Figure Legend text within the main article file. The Abstract Figure is a piece of artwork designed to give readers an immediate understanding of the research and should summarise the main conclusions. If possible, the image should be easily 'readable' from left to right or top to bottom. It should show the physiological relevance of the manuscript so readers can assess the importance and content of its findings. Abstract Figures should not merely recapitulate other figures in the manuscript. Please try to keep the diagram as simple as possible and without superfluous information that may distract from the main conclusion(s). Abstract Figures must be provided by authors no later than the revised manuscript stage and should be uploaded as a separate file during online submission labelled as File Type 'Abstract Figure'. Please also ensure that you include the figure legend in the main article file. All Abstract Figures should be created using BioRender. Authors should use The Journal's premium BioRender account to export high-resolution images. Details on how to use and access the premium account are included as part of this email.

- Please include a full title page as part of your main article (Word) file, which should contain the following: title, authors, affiliations, corresponding author name and contact details, keywords, and running title.

Reviewing Editor's Comments:

This study investigates how C-low threshold mechanoreceptors (C-LTMRs) in the hairy skin of mouse hind limbs respond to different types of mechanical brush stimuli. To examine responses, the authors measure calcium signals in genetically modified mice in which GCamp is expressed specifically in C-LTMRs. Besides establishing a new high throughput method to examine the responsiveness of C-LTMRs to different stimuli, the study also includes one main novel finding, which is that C-LTMRs are more efficiently activated by stroking hair against the direction of hair growth rather than in the same direction (i.e., the opposite direction from when petting an animal). This goes against a long-standing idea that has been proposed without direct experimental evidence that activity in C-LTMRs is associated with pleasant sensation. This study has been reviewed by two expert reviewers, who were generally enthusiastic about the work, believing that the studies were well-done and the results interesting. A recent, more broadly focused study used similar genetic methods to analyze the responsiveness of a range of DRG somatosensory neuron subtypes, including C-LTMRs, but that study did not examine the direction selectivity of C-LTMRs as was done here. There were however some concerns, which include the ones below. These and all other points brought up by the reviewers will need to be addressed. The main concerns are:

1. Figure 1 should be included as a Supplemental result rather than a main Figure. The figure describes a viral method to obtain expression of GCamp that the authors do not end up using in their study.
2. The authors should expand the analysis of the calcium signals to including the integrated response (not just peak). It would be important to know that the main findings of the study around the direction-selectivity of the neural response hold if this method of quantification is used as well as with the peak response.
3. As described by Reviewer 2, some limitations of their experiments should be better explained. These include complexities in the classifying the neurons as C-LTMRs versus C-mechanoreceptors, as well as limitations in using punctate stimuli as a method to map receptive fields.
4. The literature should be more critically evaluated, as outlined by Reviewer 2.
5. There are a number of details in the animal procedures that are not clear. These include methods of euthanasia and terminal procedures and, potentially, analgesia and post-operative care. The reviewing editor had trouble determining whether the mice were immediately euthanized after the in vivo microscopy (most likely) versus maintained alive.
6. Exact p-values should generally be provided (see Author Guidelines).
7. Standard deviations rather than standard errors should be provided.

8. The last paragraph of the Methods should provide a full description of the statistical methods that were used (see Author Guidelines).

Senior Editor's Comments:

The Journal of Physiology does not have supplemental material section, all relevant data/figures should be incorporated to the main article. Individual datapoints on current Suppl. Fig. 2 should be shown.

Referee #1:

The study by Evangelia Semizoglou and colleagues investigates how C low threshold mechanoreceptors (C-LTMRs) in the hairy skin of mouse hind limbs respond to different types of mechanical stimuli. The authors used genetically encoded calcium sensor targeted in C-LTMRs using a conditional Cre dependent strategy with the TH-2a-CreERT2 driver line and a GCamp Cre dependent reporter line to measure the activity of these neurons in response to controlled skin dynamic brushing or punctate static mechanical stimulations. The study found that C-LTMRs are more efficiently activated by skin stroking against the hair rather than in the direction of hair growth, which is counterintuitive given the supposed pleasantness of stroking in the hair growth direction. The study is well-executed, and the results are clear. This study complements a similar but much more extended study recently published by Qi et al. (Cell 2024 <https://doi.org/10.1016/j.cell.2024.02.006>), which also used intracellular calcium imaging in vivo as a proxy of neuronal activation of nearly all classes of primary afferent neurons, including C-LTMRs. The report is interesting, but it lacks a mechanistic hypothesis regarding the directional C-LTMR sensitivity. One additional point to note is that the description of the negative results with the viral approach in the TH-2a-CreERT2 in Figure 1 does not add an important point to the study. This may be more relevant in a supplementary figure and just a mention in the text.

Referee #2:

Review Semizoglou et al

In this paper the authors have established a calcium imaging protocol to examine the responses of putative C-LTMRs with cell bodies in the L4 ganglion to brush stimuli in a higher throughput fashion than conventional single fiber recording methods. A key finding from the present study that has not been noticed in a multitude of previous studies is that there is a crude direction sensitivity for C-LTMRs in the mouse hind limb being much more prominently activated by hair movement against the grain (like stroking a dog backwards, not nice). This main finding is more or less at odds with the pleasant touch hypothesis which has been tirelessly promoted by many researchers with little direct evidence. The paper is well done and

the experiments are technically demanding and the data described clearly. I have some points that the authors should consider in a revision.

Major points:

After reading the methods it is still unclear what is measured as a response. In the figures the calcium signal is described as $\max \Delta F/F$ which suggests that just the peak signal was measured. However, it would seem to make more sense to measure the integrated response (area under the curve) as this may be a better reflection of the number of action potentials fired.

The comparison between punctate stimuli is important but also very tricky. One thing that is striking to me is that many of the so-called C-LTMRs actually respond to arrange of von Frey stimuli that easily activated classical nociceptors. This should be admitted by the authors as it is in my opinion really unclear whether a C-LTMR is actually only responding to hair movement. I think the literature does not disprove the quite likely case that many classical nociceptors may be responsive to hair movement and could be classified by one investigator as a C-LTMR and another as a C-mechanoreceptor depending on the stimuli used.

One technical issue with the punctate stimulus experiment is that it is not clear if with this method the precise RF location of individual cells can ever be mapped with any confidence. It is as far as I can see an offline method. Thus, if an afferent activated by reverse hair movement had a small punctate RF with a mechanical threshold of a nociceptor it may never be located with this method. These limitations should be discussed. In general, the Discussion is very focused on the pleasant touch hypothesis, which remains for me just that a hypothesis. The authors should deal with the literature a bit more critically.

I can give some examples.

The paper by David Anderson in Nature (ref 17) should be deleted and not discussed. This paper although appearing in Nature actually showed the opposite of what is in the title, and is in my opinion a nonsense paper that should not be quoted in the literature. The authors could not activate the MRGPRB4+ neurons with any mechanical stimuli, never mind brush (data is in the supplementary material). The paper is entirely focused on calcium imaging of synapse in the spinal cord that are probably only activated indirectly by primary afferent depolarization.

Seal et al (ref 12) was essentially retracted in later papers. The paper in no way shows that C-LTMRs drive neuropathic pain. I would strongly suggest deleting this reference.

Francois et al (ref 26) is cited for showing that slow movements activate C-LTMRs did not actually show this. They show recording from putative C-LTMRs responding to ramp and hold stimuli, not stimuli with varying velocities. Although this paper is interesting it is also misleading the authors claim that deletion of Cav3.2 reduces the sensitivity of C-LTMRs, but Cav3.2 is highest expression is in D-hair mechanoreceptors where its deletion severely reduces the sensitivity of these neurons to slowly moving stimuli (PMID 21486775 not quoted). If the authors quote Francois et al they should put it into this context.

Minor comments:

von Frey filaments were invented by Maximilian von Frey. Therefore they are von Frey filaments

END OF COMMENTS

Reviewing Editor's Comments:

This study investigates how C-low threshold mechanoreceptors (C-LTMRs) in the hairy skin of mouse hind limbs respond to different types of mechanical brush stimuli. To examine responses, the authors measure calcium signals in genetically modified mice in which GCamp is expressed specifically in C-LTMRs. Besides establishing a new high throughput method to examine the responsiveness of C-LTMRs to different stimuli, the study also includes one main novel finding, which is that C-LTMRs are more efficiently activated by stroking hair against the direction of hair growth rather than in the same direction (i.e., the opposite direction from when petting an animal). This goes against a long-standing idea that has been proposed without direct experimental evidence that activity in C-LTMRs is associated with pleasant sensation. This study has been reviewed by two expert reviewers, who were generally enthusiastic about the work, believing that the studies were well-done and the results interesting. A recent, more broadly focused study used similar genetic methods to analyze the responsiveness of a range of DRG somatosensory neuron subtypes, including C-LTMRs, but that study did not examine the direction selectivity of C-LTMRs as was done here. There were however some concerns, which include the ones below. These and all other points brought up by the reviewers will need to be addressed. The main concerns are:

1. Figure 1 should be included as a Supplemental result rather than a main Figure. The figure describes a viral method to obtain expression of GCamp that the authors do not end up using in their study.

We appreciate the careful consideration of the figures in our manuscript. Although we believe that the findings in Fig 1 are important for the readers of the Journal of Physiology we would be willing to move these into supplemental figures if required (as also mentioned by reviewer #1) However, the senior editors comments suggest that the Journal of Physiology does not have a supplemental figure section. We will follow the

advice from the senior editor, and have left the figure in the main text, however, please would you be able to advise on this if an alternative is preferred.

2. The authors should expand the analysis of the calcium signals to including the integrated response (not just peak). It would be important to know that the main findings of the study around the direction-selectivity of the neural response hold if this method of quantification is used as well as with the peak response.

We very much appreciate the careful review of our methodology and have ourselves spent quite a long time thinking about his problem. Although more intuitive, the area under the curve analysis artificially makes the effect of brush speed more pronounced, as the width of the response decreases when the speed and frequency of the stimulus increase. This was the reason for the initial choice of peak response.

However, as the reviewer mentioned, it would be good to ensure that the results are robust in the face of different analytical approaches. Therefore, we have included the AUC calculations with the above-mentioned caveat included in the results text.

3. As described by Reviewer 2, some limitations of their experiments should be better explained. These include complexities in the classifying the neurons as C-LTMRs versus C-mechanoreceptors, as well as limitations in using punctate stimuli as a method to map receptive fields.

We will address these concerns more carefully in the manuscript, please refer to the comments made by the reviewer, where we have addressed this in detail.

4. The literature should be more critically evaluated, as outlined by Reviewer 2.

This too will be discussed at detail in the response to reviewer 2

5. There are a number of details in the animal procedures that are not clear. These include methods of euthanasia and terminal procedures and, potentially, analgesia and post-operative care. The reviewing editor had trouble determining whether the mice were immediately euthanized after the in vivo microscopy (most likely) versus maintained alive.

We apologise for this oversight! We have now included details on the method of euthanasia. No analgesia or post-operative care was given as the experiments and the anaesthesia used were terminal.

6. Exact p-values should generally be provided (see Author Guidelines).

We apologize for this oversight. The p-values in question are part of a figure legend where, if acceptable, we will retain approximate p-value comparisons for ease of assessment. However, we have now referenced the text, which includes the exact p-values, in the figure legend to ensure readers can easily locate the exact p-values.

7. Standard deviations rather than standard errors should be provided.

We have now updated the figures to display standard deviation, as opposed to standard errors.

8. The last paragraph of the Methods should provide a full description of the statistical methods that were used (see Author Guidelines).

We have expanded on the statistical methods that were used and included this in the last paragraph of the methods section.

Senior Editor's Comments:

The Journal of Physiology does not have supplemental material section, all

relevant data/figures should be incorporated to the main article. Individual datapoints on current Suppl. Fig. 2 should be shown.

Reviewer #1 and the reviewing editor asked us to move the current Fig. 1 into the supplemental. This is at odds to this comment. Based on your comment, we have therefore decided to remove all supplemental data or incorporate it into the main figures/text. We have also now shown individual data points for Suppl. Fig 2 (now located in Fig 4 H&I). We are happy to revise this based on any additional editorial guidance.

Referee #1:

The study by Evangelia Semizoglou and colleagues investigates how C low threshold mechanoreceptors (C-LTMRs) in the hairy skin of mouse hind limbs respond to different types of mechanical stimuli. The authors used genetically encoded calcium sensor targeted in C-LTMRs using a conditional Cre dependent strategy with the TH-2a-CreERT2 driver line and a GCamp Cre dependent reporter line to measure the activity of these neurons in response to controlled skin dynamic brushing or punctate static mechanical stimulations. The study found that C-LTMRs are more efficiently activated by skin stroking against the hair rather than in the direction of hair growth, which is counterintuitive given the supposed pleasantness of stroking in the hair growth direction. The study is well-executed, and the results are clear. This study complements a similar but much more extended study recently published by Qi et al. (Cell 2024 <https://doi.org/10.1016/j.cell.2024.02.006>), which also used intracellular calcium imaging in vivo as a proxy of neuronal activation of nearly all classes of primary afferent neurons, including C-LTMRs. The report is interesting, but it lacks a mechanistic hypothesis regarding the directional C-LTMR sensitivity. One additional point to note is that the description of

the negative results with the viral approach in the TH-2a-CreERT2 in Figure 1 does not add an important point to the study. This may be more relevant in a supplementary figure and just a mention in the text.

We thank the reviewer for the very encouraging summary of our work. We appreciate and concede that we currently do not hold any data for a mechanistic hypothesis as unfortunately this was not within the scope of the work we were conducting. However, we have made some suggestions as to possible mechanisms based on previous literature, and we hope that this will be an active research interest in the future.

Referring to your point about figure 1, while moving Fig 1 to supplementary figure is a valid suggestion, we have been advised by the editorial team that JPhys does not support supplementary data and that it should all be included in the main text. We feel we have included this data at the most appropriate place in the manuscript and we believe that it is an important (negative) finding which will be of interest to other colleagues who are trying to virally target C-LTMRs. We hope to spread awareness of this finding, so colleagues do not waste time and animals on suboptimal experimental approaches.

Referee #2:

Review Semizoglou et al

In this paper the authors have established a calcium imaging protocol to examine the responses of putative C-LTMRs with cell bodies in the L4 ganglion to brush stimuli in a higher throughput fashion than conventional single fiber recording methods. A key finding from the present study that has not been noticed in a multitude of previous studies is that there is a crude direction sensitivity for C-LTMRs in the mouse hind limb being much more prominently activated by hair movement against the grain (like

stroking a dog backwards, not nice). This main finding is more or less at odds with the pleasant touch hypothesis which has been tirelessly promoted by many researchers with little direct evidence. The paper is well done and the experiments are technically demanding and the data described clearly. I have some points that the authors should consider in a revision.

We thank the reviewer for their encouraging summary, and we are pleased that our manuscript is well received.

Major points:

After reading the methods it is still unclear what is measured as a response. In the figures the calcium signal is described as $\max \Delta F/F$ which suggest that just the peak signal was measured. However, it would seem to make more sense to measure the integrated response (area under the curve) as this may be a better reflection of the number of action potentials fired.

We thank the reviewer for their detailed assessment of our approaches. We had similar concerns and have spent some time discussing this amongst ourselves. The data can be analysed in a multitude of ways, we chose to analyse the peak response as we believe it is the most informative. We do agree that under different experimental condition, measuring the AUC can be more reflective of the number of APs fired. However, we have detailed above that analysing AUC would lead to confounds. In an attempt to avoid these confounds we originally decided that the maximal response would be the least biased while still reflecting the overall neuronal activity. However, we accept the reviewer's comments that the results should be robust in the face of different analytical approaches. Therefore, we have included the AUC analysis while mentioning the above caveat in the text to maintain analytical rigour and show robustness.

The comparison between punctate stimuli is important but also very tricky. One thing that is striking to me is that many of the so-called C-LTMRs

actually respond to arrange of von Frey stimuli that easily activated classical nociceptors. This should be admitted by the authors as it is in my opinion really unclear whether a C-LTMR is actually only responding to hair movement. I think the literature does not disprove the quite likely case that many classical nociceptors may be responsive to hair movement and could be classified by one investigator as a C-LTMR and another as a C-mechanonociceptor depending on the stimuli used.

We agree that this is a difficult aspect to fully resolve. We agree that C-LTMRs do appear to respond to a range of mechanical stimuli, even stimuli that are suprathreshold (and in the range of C-mechanonociceptors), which cannot exclude a role for them during high-mechanical stimulation. We think the key to distinguishing them relies on the responsiveness to multiple stimuli, for example responding to brush and low-threshold punctate mechanical stimuli. To that point we consider 0.07g (0.68mN), 0.16g (1.57mN) and 0.4g (3.92mN) all low threshold stimuli. In the literature, most C-nociceptors usually do not respond to these forces but to much higher forces, but it is true that threshold responders of C-nociceptors will form a bell curve distribution with a few showing thresholds leftward of the average. This, as the reviewer suggests brings into question investigator dependent classification issues.

However, we believe that the strength of our study is the genetic targeting of GCAMP solely to the TH population. Our data (and others) demonstrates that the majority if not all of this genetically defined population is mechanosensitive to low forces, and responds to brush, so therefore we believe the term C-LTMR defines them well. Studies have also shown that chemogenetic activation of C-LTMRs, induces a condition place preference, rather than an avoidance (Hazard et al., 2022, ref 34 in our updated manuscript). Lending some support that this genetically labelled population are not necessarily classical nociceptors that are "miss-defined". We have also learnt a lot from human studies. Using human microneurography C-LTMRs are defined by their unique characteristics (that are not shared with C-nociceptors). C-LTMRs do not show activity dependent slowing, they are

responsive to brush (velocity dependent), and respond to punctate mechanical stimuli. Their thresholds to punctate stimuli are low, but it is often underappreciated that they can also encode stronger mechanical stimuli too (supplementary figure 4, Middleton et al. 2022). We believe that this suggests C-LTMRs may have a broader and yet unidentified role in sensory physiology, in addition and beyond, low-threshold detection. Finally, are rodent TH positive C-LTMRs the direct species comparator to human C-LTMRs?, it is likely we do not yet have a complete picture, but we do believe with our current knowledge and comparing their neurophysiological responses to different stimuli, they demonstrate striking similarities.

One technical issue with the punctate stimulus experiment is that it is not clear if with this method the precise RF location of individual cells can ever be mapped with any confidence. It is as far as I can see an offline method. Thus, if an afferent activated by reverse hair movement had a small punctate RF with a mechanical threshold of a nociceptor it may never be located with this method. These limitations should be discussed. In general, the Discussion is very focused on the pleasant touch hypothesis, which remains for me just that a hypothesis.

We appreciate this alternative viewpoint and would like to clarify some of the technical limitations we encountered when measuring the von Frey response.

Firstly, our approach is not a precise method for receptive field mapping, we have not collected information on receptive field size or shape. While certain types of in vivo calcium imaging could technically be used for receptive field mapping one notable strength of this method is its ability to visualize multiple cells simultaneously—hundreds, when using non-specific labelling of the DRG. However, this requires also the stimulation of hundreds of receptive fields concurrently, in this case prohibiting receptive field mapping. Certainty when applying brush stimuli we were unable to map receptive fields to responding cells.

As suggested by the reviewer, we were able to match cells that showed responses to both brush and von Frey stimuli offline by analysing the response traces. It follows, that when we assess the probability of a von Frey response given a brush response and vice versa, we are not evaluating the overall overlap between dynamic and punctate stimulus responses, as we cannot ensure the stimulation of the same receptive fields. Instead, we assess whether there is for example a relative difference in the probability of a von Frey response, depending on whether the cell previously responded to reverse vs forward brushing (and vice versa for von Frey filament strength). This has been described in the results section

The authors should deal with the literature a bit more critically. The paper by David Anderson in Nature (ref 17) should be deleted and not discussed. This paper although appearing in Nature actually showed the opposite of what is in the title, and is my opinion a nonsense paper that should not be quoted in the literature. The authors could not activate the MRGPRB4+ neurons with any mechanical stimuli, never mind brush (data is in the supplementary material). The paper is entirely focused on calcium imaging of synapse in the spinal cord that are probably only activated indirectly by primary afferent depolarization.

We are delighted that the reviewer shares our opinion and concerns on ref 17, we strongly believe that the MRGPRB4 population are not C-LTMRs. We very conservatively included the reference, as we are so often asked to quote it. We have removed the reference and its discussion.

Seal et al (ref 12) was essentially retracted in later papers. The paper is no way shows that C-LTMRs drive neuropathic pain. I would strongly suggest deleting this reference.

We absolutely agree with the reviewer and are aware of the limitations of the cited paper by Seal et al. 2009. A subsequent study from the same group dissected the role of vGLUT3 in more detail and identified a key role of vGLUT3 in spinal neurons driving aspects of neuropathic pain. However,

this does not alter that key point that we intended to make, which is vGLUT3 (in the DRG) is a marker of C-LTMRs. We believe that this is important to mention, but we are not set on citing Seal et al 2009, so instead we have replaced that reference with a paper that uses vGLUT3 as an anatomical marker of C-LTMRs to assess their synaptic organisation (Larsson & Broman 2019m). We hope the reviewer accepts this compromise.

Francois et al (ref 26) is cited for showing that slow movements activate C-LTMRs did not actually show this. They show recording from putative C-LTMRs responding to ramp and hold stimuli, not stimuli with varying velocities. Although this paper is interesting it is also misleading the authors claim that deletion of Cav3.2 reduces the sensitivity of C-LTMRs, but Cav3.2 is highest expression is in D-hair mechanoreceptors where its deletion severely reduces the sensitivity of these neurons to slowly moving stimuli (PMID 21486775 not quoted). If the authors quote Francois et al they should put it into this context.

We thank the reviewer for spotting this. We appreciate the very important and primary role of Cav3.2 in D-Hair primary afferents. However, Cav3.2 is also expressed in C-LTMRs, and Francios et al selectively delete Cav3.2 from Nav1.8 expressing sensory neurons only (which would not include D-hairs). Importantly, they show in the supplementary data that Cav3.2 currents are unchanged in medium/large A-delta sensory neurons when using their KO model. We agree that these specifics should be clarified for the reader, however, we do concede that we have used the incorrect reference for the statement we are making. Francios did not show that moving stimuli activate the C-LTMRs they record from. The statement we make is that C-LTMR activity depends on stimulus movement, which has also been observed in mammals. We have removed ref.26 and instead included Kumazawa & Perl (1977) and Zotterman (1939)

We very much appreciate the expert input from the reviewer and we have made every effort to update our referencing in the discussion of the

literature! We hope to have addressed all the concerns with the papers mentioned in our discussion and we believe it is now a far superior review and discussion of the proceeding literature!

Minor comments:

von Frey filaments were invented by Maximilian von Frey. Therefore they are von Frey filaments

We apologise for the oversight. This has now been adjusted

Dear Dr Chisholm,

Re: JP-RP-2024-286631R1 "In vivo calcium imaging reveals directional sensitivity of C-low threshold mechanoreceptors" by Evangelia Semizoglou, Laure Lo-Re, Steven J Middleton, Jimena Perez-Sanchez, Tommaso Tufarelli, David L Bennett, and Kim I Chisholm

Thank you for submitting your manuscript to The Journal of Physiology. It has been assessed by a Reviewing Editor and by 2 expert referees and we are pleased to tell you that it is acceptable for publication following satisfactory revision.

REVISION CHECKLIST:

We look forward to receiving your revised submission.

Yours sincerely,

Katalin Toth
Senior Editor
The Journal of Physiology

Reviewing Editor's comments:

Your revised manuscript has been reviewed by two expert reviewers who both felt that you did a nice job of addressing prior concerns. You also adequately addressed the points raised about animal use and statistics. Around the question of the potential supplementary figure, that was my prior error. You should follow the guidance of the senior editor and maintain the figure as a regular figure. Reviewer 2 did raise a few very minor points that will still need to be addressed.

Referee #1:

The revised version of the manuscript addresses all the concerns that I had. This is a solid study reporting an original observation. This is of general interest to the field of touch physiology. The conclusions are supported by the experiments and the discussion is adequate in the context of the literature.

Referee #2:

The authors have done a good job answering both referees comments. Some minor points should be answered before publication.

Formatting page 10 is amiss.

Page 15 CLTMR written instead of C-LTMR. This may be a more common occurrence the authors should check...

line 417 unclear what is meant by "inconsistent results" be more specific please.

END OF COMMENTS

Reviewing Editor's comments:

Your revised manuscript has been reviewed by two expert reviewers who both felt that you did a nice job of addressing prior concerns. You also adequately addressed the points raised about animal use and statistics. Around the question of the potential supplementary figure, that was my prior error. You should follow the guidance of the senior editor and maintain the figure as a regular figure. Reviewer 2 did raise a few very minor points that will still need to be addressed.

Thank you for taking the time to review our manuscript. We appreciate your careful consideration and valuable feedback. We will revise the manuscript according to Reviewer 2's suggestions and retain the figure in the main document as initially presented.

Referee #1:

The revised version of the manuscript addresses all the concerns that I had. This is a solid study reporting an original observation. This is of general interest to the field of touch physiology. The conclusions are supported by the experiments and the discussion is adequate in the context of the literature.

Thank you very much, we appreciate your help in reviewing our manuscript!

Referee #2:

The authors have done a good job answering both referees comments. Some minor points should be answered before publication.

Thank you very much for your thoughtful consideration of our manuscript. We will address the points raised below

Formatting page 10 is amiss.

This has now been adjusted

Page 15 CLTMR written instead of C-LTMR. This may be a more common occurrence the authors should check...

This has been changed in the manuscript

line 417 unclear what is meant by "inconsistent results" be more specific please.

This has been changed to "no longer able to brush at a continuous speed below 0.5cm/s" as the brush would sort of "stutter" or even stop altogether below 0.5cm/s.

Dear Dr Chisholm,

Re: JP-RP-2024-286631R2 "In vivo calcium imaging reveals directional sensitivity of C-low threshold mechanoreceptors" by Evangelia Semizoglou, Laure Lo Re, Steven J Middleton, Jimena Perez-Sanchez, Tommaso Tufarelli, David L Bennett, and Kim I Chisholm

We are pleased to tell you that your paper has been accepted for publication in The Journal of Physiology.

*****IMPORTANT*****

We note in your cover letter that you expressed interest in being jointly published with another submission from Andrew Marshall and colleagues.

The Marshall paper is still under review/revision and we do not have a clear timeline yet of when it might be accepted. We just wanted to check with you whether you wanted us to hold issue publication of your article above for several weeks (or potentially months) until we have a final decision on the Marshall paper?

We will hold your paper from production until you let us know (as this has implications for the instructions we will need to issue to our publisher).

Once your paper enters production, it will be published online within a couple of weeks, even if we hold it from issue allocation.

We look forward to hearing from you about whether you want to wait to be published in an issue with the Marshall paper (assuming it is accepted) in due course.

Yours sincerely,

Katalin Toth
Senior Editor
The Journal of Physiology

If you would like to receive our 'Research Roundup', a monthly newsletter highlighting the cutting-edge research published in The Physiological Society's family of journals (The Journal of Physiology, Experimental Physiology, Physiological Reports, The Journal of Nutritional Physiology and The Journal of Precision Medicine: Health and Disease), please click this link, fill in your name and email address and select 'Research Roundup':
<https://www.physoc.org/journals-and-media/membernews>

• **TRANSPARENT PEER REVIEW POLICY:** To improve the transparency of its peer review process, The Journal of Physiology publishes online as supporting information the peer review history of all articles accepted for publication. Readers will have access to decision letters, including Editors' comments and referee reports, for each version of the manuscript as well as any author responses to peer review comments. Referees can decide whether or not they wish to be named on the peer review history document.

• You can help your research get the attention it deserves! Check out Wiley's free Promotion Guide for best-practice recommendations for promoting your work at: www.wileyauthors.com/eeo/guide. You can learn more about Wiley Editing Services which offers professional video, design, and writing services to create shareable video abstracts, infographics, conference posters, lay summaries, and research news stories for your research at: www.wileyauthors.com/eeo/promotion.

• **IMPORTANT NOTICE ABOUT OPEN ACCESS:** To assist authors whose funding agencies mandate public access to

published research findings sooner than 12 months after publication, The Journal of Physiology allows authors to pay an Open Access (OA) fee to have their papers made freely available immediately on publication.

EDITOR COMMENTS

Reviewing Editor:

The changes you have made in your revised manuscript addresses all of the remaining concerns.